# Identification of Sieve Element Occlusion Gene (SEOs) Family in Rubber Trees (*Hevea brasiliensis* Muell. Arg.) Provides Insights to the Mechanism of Laticifer Plugging

**Xuan Wang** [1,2,†]**, Linlin Cheng** [2,†]**, Wentao Peng** [2]**, Guishui Xie** [2]**, Zifan Liu** [1,*] **and Feng An** [2,*]

1    College of Tropical Crops, Hainan University, Haikou 570228, China; leahdou11@163.com
2    Hainan Danzhou Agro-Ecosystem National Observation and Research Station, Rubber Research Institute, Chinese Academy of Tropical Agricultural Sciences, Danzhou 571737, China; chenglinlin2004@163.com (L.C.); dawentao@163.com (W.P.); xie23300459@163.com (G.X.)
\*    Correspondence: jiangxilaobiao@163.com (Z.L.); an-f@catas.cn (F.A.)
†    These authors contributed equally to the work.

**Abstract:** P proteins encoded by *SEOs* (sieve element occlusion) have been shown to be associated with the blockage of sieve tubes after injury in many plants, but the presence of *SEO* genes and their association with rubber tree laticifer plugging and latex yield remain unclear. Through a systematic identification and analysis, seven *SEO* genes were identified from the rubber tree genome. The physicochemical properties of their proteins, gene structures, conserved domains, and locations on chromosomes were analyzed. According to their phylogenetic distance, *HbSEOs* were divided into two clusters. The transcript levels of *HbSEO* genes varied with tissues, in which *HbSEO3* and *HbSEO4* were most highly expressed in leaf, bark, and latex. *HbSEOs* could be induced by ethephon, methyl jasmonate, mechanical injury, and tapping; furthermore, they were highly expressed in trees with short flow duration, suggesting their possible association with rubber tree laticifer plugging and latex yield. To our knowledge, this is the first report of *HbSEOs* in rubber trees. It provides us with a better understanding of the mechanism of laticifer plugging.

**Keywords:** rubber tree; sieve element occlusion; laticifer plugging; gene structure; gene expression

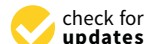



## 1. Introduction

Natural rubber is an important industrial raw material, which comes from the cytoplasm flow of rubber tree phloem after periodic tapping [1,2]. The rubber tree latex yield after each tapping of rubber trees is determined by latex flow velocity and flow duration [3]. However, both latex flow duration and flow velocity are relevant to laticifer plugging, since laticifer plugging after tapping forms gradually, resulting in the slowing down and eventually total stoppage of the latex flow [4].

The formation of laticifer blockage at the tapping cut of rubber trees has been confirmed by both repeated tapping and bark anatomy observation [5]. Many hypotheses have been suggested to explain the mechanism. In the 'electro-neutralization' hypothesis, 'enzyme' hypothesis, and 'lectin' hypothesis, it is suggested that the rubber particles play a major role in laticifer plugging [6]. Nevertheless, in later research, it was found the rubber particles remained intact and did not coagulate when the laticifers had recently stopped draining. The 'protein network' mechanism was, therefore, proposed [7]. In this mechanism, an electron-dense protein network is formed immediately after tapping. Then, the 'protein network' becomes denser and denser with the increase of the draining time and the accumulation of rubber particles near the 'mesh', and a 'plug' is gradually formed, causing the eventual blockage of the severed laticifers. Nevertheless, there are still controversies about the composition, agglomeration mechanism, and inducing factors of the plugging. Wu and Hao [3,4] found that there were fiber structure and tubular structure

P-proteins in the secondary phloem of young rubber branches. The conformational change of P-protein, from fibrous to tubular, after mechanical injury was associated with the plugging of the sieve tube after wounding [5]. However, there have been no in-depth studies on the P-protein coding genes and their relationship with laticifer blockage.

The *SEO* (sieve element occlusion) gene was first found in leguminous plants as an encoding structural component of forisomes. Subsequently, it was found that *SEO* genes are widely distributed in non-legume angiosperms, to encode P-proteins having similar ultra-structure and functions with legume isoforms [8]. In most angiosperm sieve elements, P-protein is a special structural protein that facilitates rapid wound healing after injury stress and prevents swelling and massive loss of photosynthates [9,10]. Local calcium concentration alterations in response to wounding cause the reversible conformational change of SEO proteins, from crystalline to dispersed state, and then the blocking of the severed sieve tubes [11]. The SEO proteins have three domains: i.e., SEO N-terminal domain (SEO-NTD), potential thioredoxin folding, and SEO C-terminal domain (SEO-CTD) [12]. However, the presence of SEOs and their relationship with rubber tree laticifer plug formation and latex yield remain unknown.

In this study, a genome-wide search was carried out to identify the *SEO* genes in rubber trees. Furthermore, the expression profile of *HbSEOs* in response to different hormone stimulation and latex flow duration was examined to investigate their potential roles in laticifer plugging and latex yield.

## 2. Materials and Methods

### 2.1. Identification of Rubber Tree SEO Genes

The whole genome, including GFF file and protein file, of the rubber tree (ASM165405v1), and all the SEO amino acid sequences for *Arabidopsis thaliana* (L.) Heynh., *Populus trichocarpa* Torr. & Gray, and *Ricinus communis* L. were downloaded from the National Center for Biotechnology Information (NCBI, https://www.ncbi.nlm.nih.gov/, accessed on 20 March 2021). Local BLASTP of homologous sequences was performed by using the SEO protein sequences of *A. thaliana*, *P. trichocarpa,* and *R. communis* as query templates and the rubber tree genome as a database. The parameters were NumofThreads: 2, E-value: 1e-5, NumofHits: 500, NumofAligns: 250; other parameters: the defaults were used during the BLASTP. The extractions of SEO amino acid sequence from rubber trees were executed using TBtools [13].

The Pfam_A HMM models were downloaded from the Pfam_database (ftp://ftp.ebi.ac.uk/pub/databases/Pfam/current_release/Pfam-A.hmm.gz, accessed on 21 March 2021), and local HMMER was screened using TBtool. Then, all the obtained HbSEO protein sequences were submitted for CDD-Search (https://www.ncbi.nlm.nih.gov/cdd/, accessed on 21 March 2021), Pfam (http://pfam.xfam.org/, accessed on 21 March 2021), and SMART (http://smart.embl-heidelberg.de/, accessed on 21 March 2021) check, to remove sequences with significant length differences and incomplete conservative domains. All SEO proteins from the rubber tree were, thus, obtained.

### 2.2. Sequence Alignment and Phylogenetic Analysis of HbSEOs

Multiple sequence alignment of SEO amino acid sequences was conducted using default parameters (Gap Opening Penalty: 10, Gap Extension Penalty: 0.10) in MEGA-X. Subsequently, the phylogenetic tree was constructed in MEGA-X software, using the maximum likelihood method with default parameters (Bootststrap Replications: 1000), and visually modified on the online website EvolView (https://www.evolgenius.info/evolview/#login, accessed on 21 March 2021).

### 2.3. Basic Informations and Sequence Analysis of the HbSEOs

HbSEO conservative domains and their gene chromosome localizations were predicted and visualized in TBtools. Secondary structures of HbSEO proteins were predicted using the NPS@:SOPMA online website tool (https://npsa-prabi.ibcp.fr/cgi-bin/npsa_automat.pl?page=npsa_sopma.html, accessed on 22 March 2021), and their basic protein properties

were collected from the ExPASy online website (https://web.expasy.org/protparam/, accessed on 22 March 2021). The sub-cellular localization was predicted with POSRT on the website (http://psort1.hgc.jp/form.html, accessed on 22 March 2021). Conserved motifs in HbSEO proteins were identified using the online tool MEME (https://meme-suite.org/meme/tools/meme, accessed on 22 March 2021), with the motif number set to 12 and the remaining parameters as default, and then displayed by TBtools. The upstream 2000 bp sequence of each *HbSEO* was extracted and submitted to the online website Plant CARE (http://bioinformatics.psb.ugent.be/webtools/plantcare/html/, accessed on 22 March 2021), to detect their cis-acting elements, such as light-responsive element, defense and stress-responsive element, low-temperature responsive element, and auxin-responsive element. Then, the cis-acting elements of *HbSEO* genes were visualized using TBtools.

*2.4. RNA Isolation and Real-Time Quantitative Analysis*

Unless otherwise mentioned, Reyan 7-33-97 tissue culture rubber tree saplings [14], which is the most widely planted and high-yield variety in Hainan, were used as plant materials. From trees with similar leaf stability, height, stem diameter, leaf number, and growth period, roots, stem tips, leaves, and barks were collected and used for RNA extraction. Meanwhile, barks and latex of mature trees were sampled from healthy rubber trees that had been regularly tapped under the 1/2S 3d taping system for about 10 years.

To apply hormone treatments, methyl jasmonate acid (MeJA, 0.005% *v/v*), salicylic acid (SA, 200 μmol/L), ethephon (ETH, 1% *v/v*), and gibberellin (GA, 3 mmol/L) were sprayed on the back of leaves of two-whirled-leaf type. The leaves were sampled at 0.5, 4, 8, 12, 24, and 48 h after treatment, with untreated leaves as control. Abrasive paper was used to induce mechanical damage at the back of the leaf, and leaf samples were collected at 0.5, 1, 2, 6, 10, 24, and 48 h after treatment. Just after sampling, the samples were quickly frozen in liquid nitrogen, and then stored in a $-80\ ^{\circ}\text{C}$ refrigerator for further use. The total RNA of each sample was extracted using a general polysaccharide polyphenol plant total RNA extraction kit, according to the manufacturer's manual (Beijing Tiangen Biotech Co., Ltd, Beijing, China). The purity and concentration of the extracted RNA was verified with an ultra-micro nucleic acid protein analyzer (NanoDrop 2000c, Thermo Scientific, Wilmington, DE, USA), to ensure the concentration of RNAs was more than 1000 ng/μL, and gel verification was used to show that RNA was not digested. After complete removal of residual gDNA in the RNA template, reverse transcription of cDNA was conducted, according to the RevertAid Strand cDNA Synthesis Kit (Thermo, Vilnius, Lithuania) instructions, and tested with agarose gel electrophoresis.

All gene-specific primers (Table 1) were designed using the Primer 3.0 software (http://frodo.wi.mit.edu/primer3/input.htm, accessed on 24 March 2021), and primer specificity was verified using Primer BLAST in NCBI and PCR amplification and agarose gel electrophoresis experiments. The rubber tree 18S rRNA gene was used as a reference gene, as it has been reported to be one of the most stably expressed reference genes in response to tapping, hormone treatment, and other experimental conditions [15]. According to the manufacturer's instruction, real-time quantitative PCR was performed using a CFX96 real-time polymerase chain reaction system (Bio-Rad, Hercules, CA, USA) SYBR Green Premix kit (Takara, Dalian, China), with the cDNA from different hormone treatments as templates. In each qRT-PCR reaction system, 10 μL 2 × SYBR Green Premix, 0.4 μL of each primer (10 μM), 1 μL cDNA (containing 100 ng of RNA template) and 8.2 μL RNase-free water were used, and the reaction program was pre-denaturation at 95 °C for 30 s, denaturation at 95 °C for 5 s, annealing extension at 60 °C for 30 s, and 40 cycles. Each sample was subjected to three biological and three technical repetitions, and the relative expression of the target gene was calculated using the $2^{-\Delta\Delta Ct}$ method, where ΔΔCt = (Ct target gene − Ct reference gene) experimental group − (Ct target gene − Ct reference gene) control group. Data Processing System software v11.0 was used for analyzing variances, following Duncan's one-way ANOVA ($p < 0.05$), and Origin 2018 (OrigionLab Cooperation, Northampton, MA, USA) was used to draw the figures.

**Table 1.** qRT-PCR primers for expression analysis of *HbSEOs*.

| Gene Accession Number (NCBI) | Rename | Forward Primer | Reverse Primer |
|---|---|---|---|
| XM_021833957.1 | *HbSEO*1 | AGGCTTGGGAGTTGTCGAGC | GCAGGACGTCAATGTTAACCCTC |
| XM_021811739.1 | *HbSEO*2 | GGCTTCCTGTTGTGGACCGAT | GGATTCACTACTTTCCCTTGCGG |
| XM_021804083.1 | *HbSEO*3 | GTGAGGCTGGAGAGCATGTG | TTCCATGACTGAAGGGCATC |
| XM_021804082.1 | *HbSEO*4 | ACGCGTTGTGTTTTGTTCATCAG | TGTGCATGGGAAGCTCCTTGT |
| XM_021811738.1 | *HbSEO*5 | GGAAGAAAGGCAACGGAATGCT | ACTCTCCATGGGACGCTTACAC |
| XM_021813449.1 | *HbSEO*6 | TAATCCAGGGATTGGCGCTG | GAAGGGTGGCACGATTCAGG |
| XM_021828364.1 | *HbSEO*7 | ACCCGTCAAGTTGAGACCAC | AGCACCAACTTAGCATCCCAGGAG |
| AB268099 | 18S rRNA | GCTCGAAGACGATCAGATACC | TTCAGCCTTGCGACCATAC |

## 3. Results

### 3.1. Identification, Classification, and Protein Property Analysis of HbSEOs

Using published SEO sequences of *A. thaliana*, *P. trichocarpa*, and *R. communis* as queries, a total of 24 rubber tree putative SEO proteins were obtained by local BLASTP and local HMMER screening of the rubber tree genome database. Then, Pfam, SMART, and CDD-Search were executed to predict the conserved regions of HbSEO. Seven HbSEO proteins containing all conserved domains (i.e., N-terminal domain, C-terminal domain and thioredoxin super-family domain) were identified and used for further study. Other proteins with partial sequences and incomplete conserve domains were excluded. The seven SEOs of rubber tree were named HbSEO1–HbSEO7.

The distribution of conserve domains for each of HbSEO proteins is shown in Figure 1. From phylogenetic analysis, the seven identified HbSEO proteins were divided into two groups, among which HbSEO3, HbSEO4, HbSEO2, HbSEO7, HbSEO6, and HbSEO1 were classed into group I, and HbSEO5 was classed into group II.

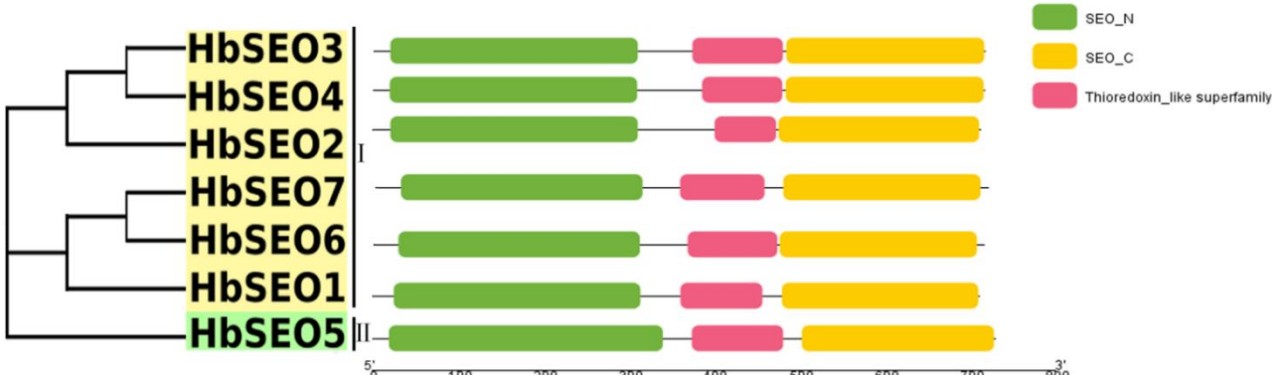

**Figure 1.** Conserved domains of HbSEO proteins. Different colors represent different domains, green represents SEO_N (SEO N-terminal domain), pink represents SEO_C (SEO N-terminal domain), and yellow represents Thioredoxin-like super-family.

The physical and chemical properties of the seven *HbSEOs* are shown in Table 2. It can be seen that the encoded proteins of the seven *HbSEOs* were between 711 (HbSEO1) and 730 (HbSEO5) amino acids, with a relative molecular weight (MW) ranging between 80.94 (HbSEO1) and 83.26 (HbSEO5) KDa, while the isoelectric point (PI) varied between 6.12 (HbSEO1, HbSEO2) and 6.29 (HbSEO5, HbSEO6). There were four stable proteins (HbSEO1, HbSEO4, HbSEO6, and HbSEO7) and three unstable proteins (HbSEO2, HbSEO3, and HbSEO5). The secondary structure of HbSEO members contained alpha helix, beta turn, and random coil, with the content: alpha helix > random coil > beta turn. Among which, HbSEO4 (56.69%) had the highest and HbSEO3 (53.34%) the lowest α-helix content; whereas, HbSEO1 (4.5%) possessed the highest and HbSEO3 (30.6%) possessed the lowest β-turn content, and HbSEO3 (32.59%) contained the most and HbSEO1 (30.38%) contained the least irregular coils. Sub-cellular localization prediction showed that HbSEO1, HbSEO3,

and HbSEO5 localized in cytoplasm, while HbSEO2 and HbSEO4 localized in mitochondria, and HbSEO6 and HbSEO7 localized in the microbody.

**Table 2.** Basic protein information of HbSEO genes.

| Gene ID | Gene Name | Genomics (bp) | CDS (bp) | Amino Acids | MW (kDa) | PI | Instability Index | α-Helix | β-Turn | Random Coil | Sub-Cellular Location |
|---|---|---|---|---|---|---|---|---|---|---|---|
| XP_021689649.1 | *HbSEO1* | 2518 | 2136 | 711 | 80.94 | 6.12 | 39.85 | 54.29% | 4.50% | 30.38% | cytoplasm |
| XP_021667431.1 | *HbSEO2* | 2463 | 2139 | 712 | 80.95 | 6.12 | 44.81 | 54.07% | 4.21% | 31.46% | mitochondria |
| XP_021659775.1 | *HbSEO3* | 2476 | 2157 | 718 | 82.14 | 6.14 | 40.56 | 53.34% | 3.06% | 32.59% | cytoplasm |
| XP_021659774.1 | *HbSEO4* | 2758 | 2157 | 718 | 81.98 | 6.13 | 37.54 | 56.69% | 4.04% | 29.81% | mitochondria |
| XP_021667430.1 | *HbSEO5* | 2485 | 2193 | 730 | 83.26 | 6.29 | 47.99 | 53.97% | 4.11% | 31.78% | cytoplasm |
| XP_021669141.1 | *HbSEO6* | 2448 | 2151 | 716 | 82.06 | 6.29 | 39.8 | 54.47% | 4.33% | 30.59% | microbody |
| XP_021684056.1 | *HbSEO7* | 2419 | 2157 | 718 | 82.36 | 6.27 | 35.64 | 55.01% | 4.32% | 30.92% | microbody |

Note: MW, molecular weight; pI, isoelectric point.

Subsequently, phylogenetic trees (Figure 2) of *HbSEOs* were constructed with another four plants, i.e., *A. thaliana*, *Manihot esculenta* Crantz, *R. communis*, and *P. trichcarpa* SEOs (Supplementary Table S1). According to their phylogenetic relationships, the 36 SEOs from the five species were grouped into three clusters, and the *HbSEOs* were again divided into two clusters, as in the results of Figure 1, using only *HbSEOs*. The SEOs of rubber tree were mostly distributed in cluster I and had a closer evolutionary distance with *M. esculenta* and *R. communis*, which all belonged to the Euphorbiaceae, as rubber trees.

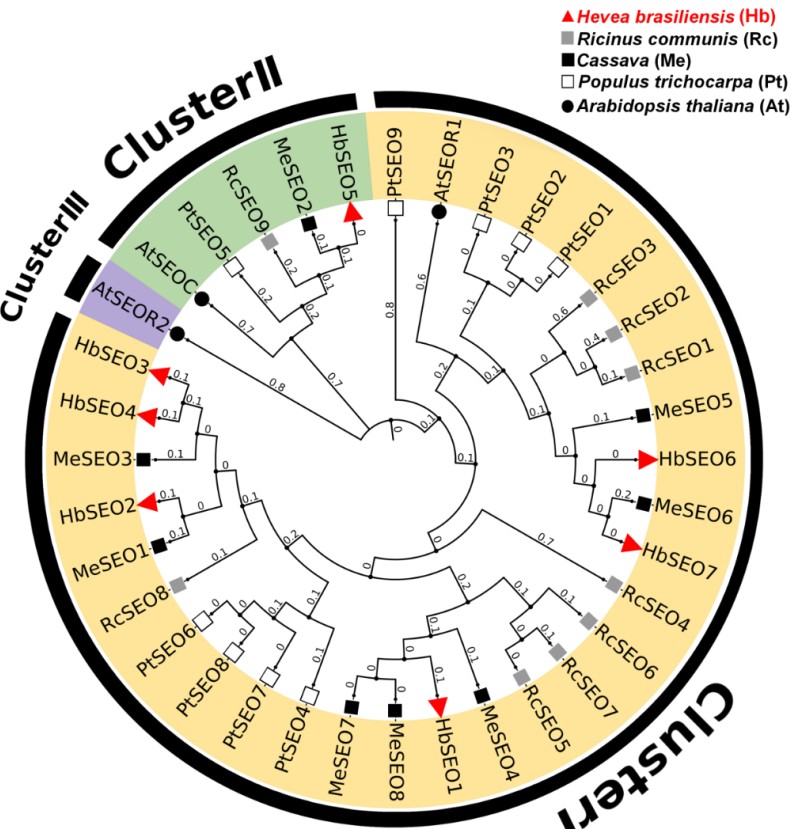

**Figure 2.** Phylogenetic analysis of SEOs from *H. brasiliensis, A. thaiana, M. esculenta, R. communis,* and *P. trichcarpa*. All complete protein sequences were aligned using MEGA-X software, and the phylogenetic tree was displayed using EvolView. The phylogenetic tree was constructed using the maximum likelihood method and JTT matrix-based model, with bootstrap replications of 1000. The numbers on branches indicate the evolutionary distance.

### 3.2. Gene Structures and Conserved Motifs of HbSEOs

Gene structure analysis showed that the *HbSEOs* were rather conservative (Figure 3). They all contained six introns and seven coding sequences. Compared to other genes, *HbSEO5* contained longer introns for the first and forth introns. More importantly, at the 3′ end of *HbSEO5*, the UTR length was much shorter than other genes. Therefore, gene length and CDS distribution were closely related with their phylogenetic relationship, making genes with similar gene structures group together.

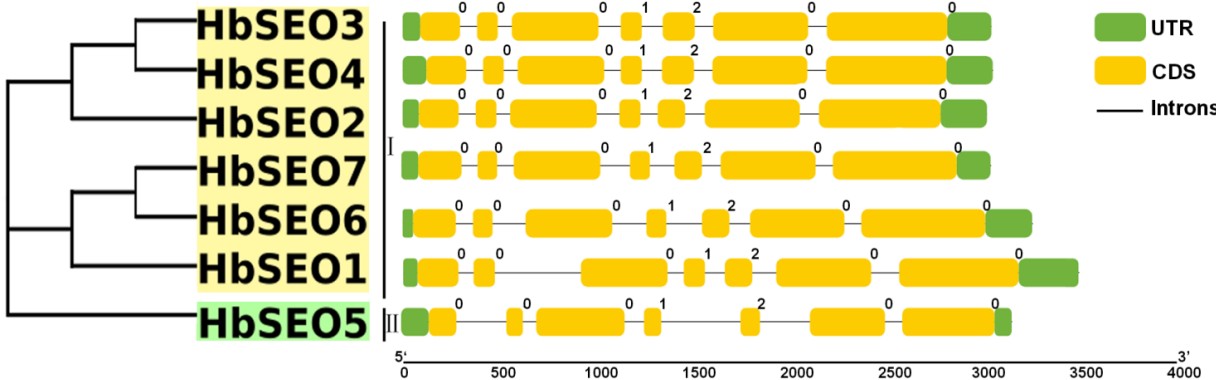

**Figure 3.** Gene structure of *HbSEOs*. The thick yellow boxes indicate CDS, the thick green boxes indicate UTR, and the black lines indicated introns.

Using MEME online tool, a total of eleven or twelve conserved motifs were identified in the HbSEO amino acid sequences (Figure 4). Most HbSEO proteins contained 12 conserved motifs, with the arrangement order being, motif 6-motif 10-motif 2-motif 3-motif 8-motif 11-motif 9-motif 1-motif 4-motif 12-motif 5-motif 7. These motifs corresponded to the three HbSEO conserved domains, with motif 6, motif 10, motif 2, motif 3, and motif 8 harbored in SEO_N; motif 11, motif 9, and motif 1 harbored in the Thioredoxin_like super-family; and motif 4, motif 12, motif 5, and motif 7 harbored in SEO_C. Nevertheless, HbSEO4 contained an extra motif 6 at a length of 300–400 and HbSEO5 lacked motif 7 at the 3′ end of the protein. The distributions of these motifs further supported the phylogenetic classification of *HbSEOs*.

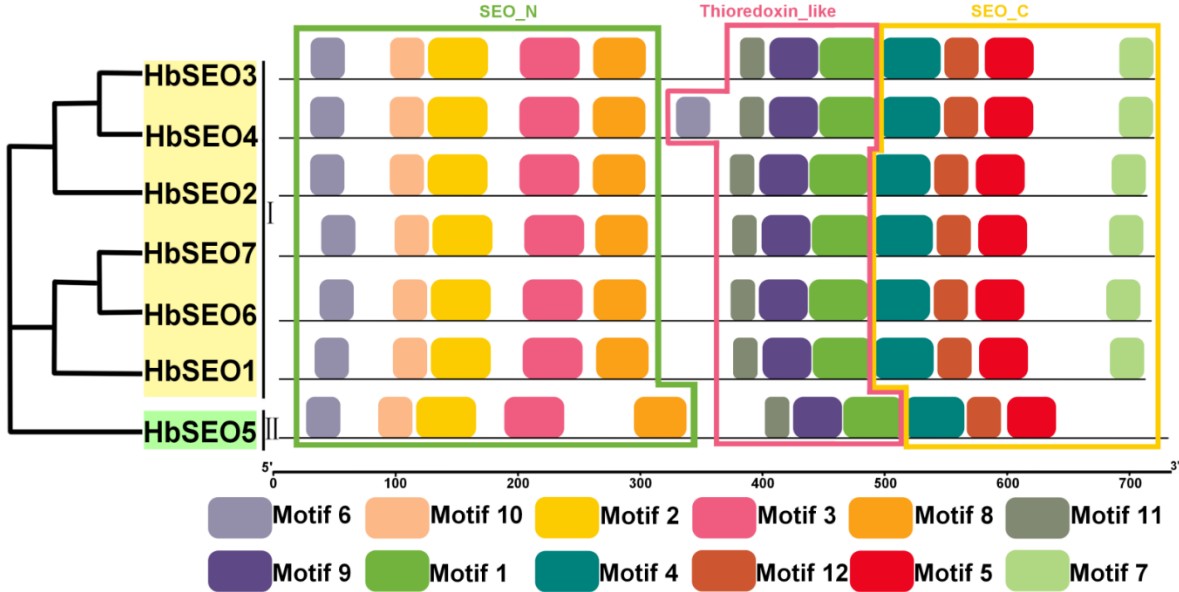

**Figure 4.** Motif analysis of HbSEO proteins. The 12 conserved motifs of the HbSEO proteins were detected using the online MEME program and drawn using TBtools. The length of protein can be estimated using the scale at the bottom.

### 3.3. Chromosome Localization

Chromosome localization analysis (Figure 5) showed that the *HbSEOs* were unevenly distributed across the five chromosomes of *H. brasiliensis*. *HbSEO2* and *HbSEO5* were located on NW_018745783.1, *HbSEO3* and *HbSEO4* were located on NW_018747589.1, and the other three genes, *HbSEO6*, *HbSEO7* and *HbSEO1*, were individually distributed on three chromosomes.

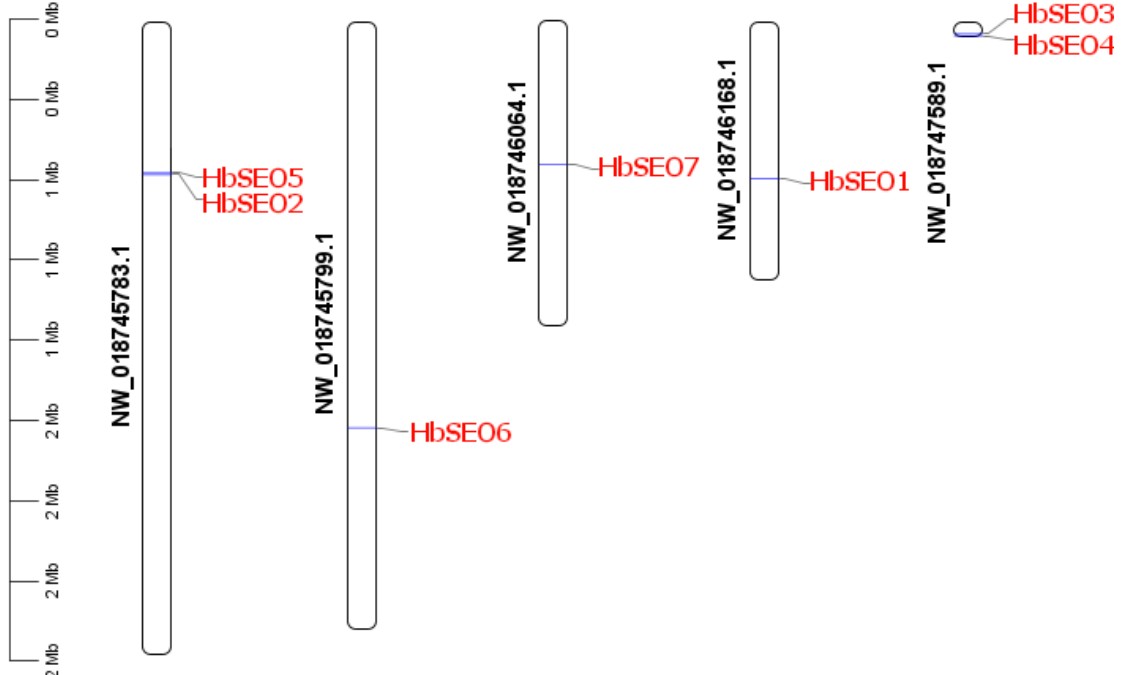

**Figure 5.** The chromosome localization of *HbSEOs*. The scale on the left side of the graph shows the length of the chromosomes in Megabase pairs (Mb). The chromosome number is indicated at the left of each chromosome.

### 3.4. Analysis of Cis-Acting Elements in Promoter Region

To predict the factors regulating *HbSEOs* from the cis-acting elements harbored in their promoter regions, 2000 bp upstream of *HbSEOs* were extracted to analyze their cis-acting elements (Figure 6), The common cis-acting elements, including those responsive to distinct plant hormones (MeJA, GA, SA, IAA, ABA, ET), stress factors (low-temperature, anaerobic, drought), seed-specific regulator, and meristem expression, were predicted. The results showed that each *HbSEOs* harbored at least one cis-element in its promoter region. The promoter regions of all seven genes contained light responsive elements and ethylene-responsive elements, implying their possible association with photosynthates utilization and ethylene induced latex yield increase. The promoter regions of *HbSEO1*, −2, −3, −6, and −7 contained MeJA response elements. Moreover, the promoter regions of *HbSEO1*, −2, −5, and −7 contained GA responsive elements, and *HbSEO1*, −2, −3, and −5 contained SA responsive elements. Meanwhile, the promoter regions of *HbSEO1*, −3, and −4 contained IAA responsive elements, and *HbSEO4*, −6, and −7 contained ABA responsive elements. While, the promoter regions of *HbSEO1*, −2, −5, and -6 contained defense and stress response elements. The existence of these cis-acting elements indicated the possibility of *HbSEOs* regulation in response to hormones such as MeJA, GA, SA, IAA, ABA, and ET. In addition, response elements related to abiotic stresses, such as low temperature, anaerobic, and drought, along with circadian control elements, seed-specific regulator element, and meristem expression response element, were also detected in their promoter regions, indicating that *HbSEO* genes might be associated with rubber tree growth, development, and a variety of abiotic stresses.

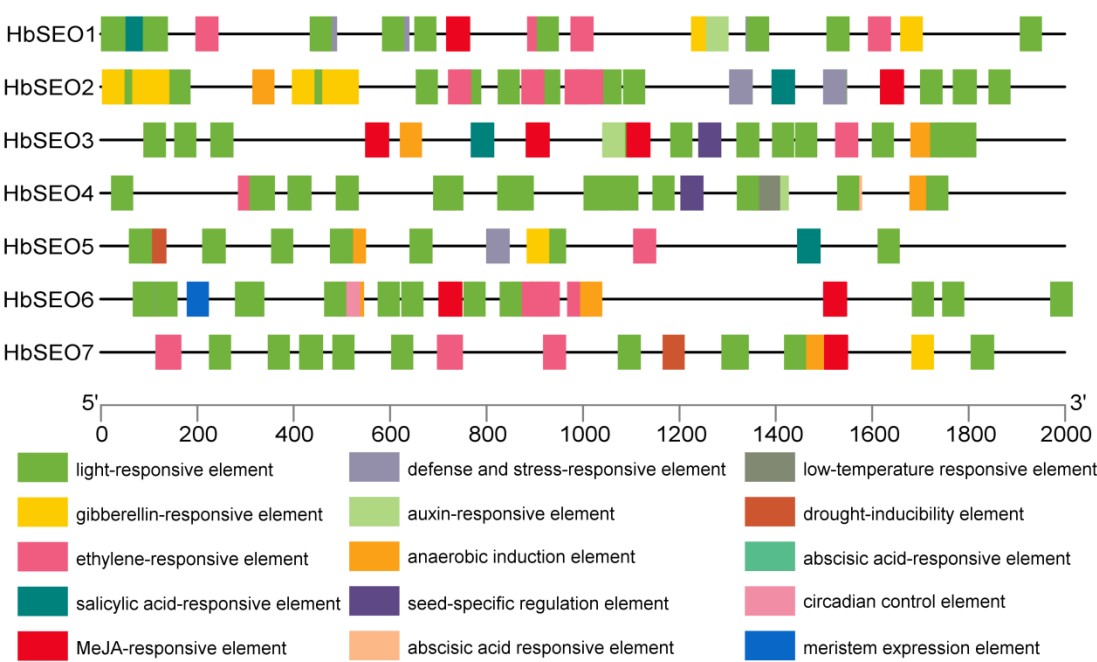

**Figure 6.** Cis-acting element analyses of *HbSEOs*. Different color boxes represent different cis-elements. The 2000 bp upstream of *HbSEO* gene promoter sequences were selected to detect promoter cis-elements using the PLANTCARE database.

### 3.5. Expression Analysis of HbSEOs

#### 3.5.1. Tissue-Specific Expressions of *HbSEOs*

Real-time fluorescence quantitative PCR (qRT-PCR) analysis of the expression of *HbSEOs* was conducted in the roots, stem tips, leaves, bark (young), and latex of rubber trees (Figure 7). Figure 7A shows that the expression levels of all *HbSEO* genes were comparatively low in leaves; *HbSEO1*, −*2*, and −*6* had the highest expression level in young stem, and *HbSEO3*, −*4*, −*5*, and −*7* were highly expressed in latex. When compared the various *HbSEOs* in the same tissues (Figure 7B–F), it was shown that in root, the transcript levels of *HbSEO2* and *HbSEO3* were the highest, and *HbSEO5* was lowest (Figure 7B); and in stem the expression of *HbSEO3* and *HbSEO4* was the highest, and *HbSEO5* was lowest (Figure 7C); while in leaf, *HbSEO3* and *HbSEO4* showed the highest, and *HbSEO6* showed the lowest, expression level (Figure 7D); however, in bark and latex, *HbSEO4* and *HbSEO3* showed the highest expression, and *HbSEO5* showed the lowest expression level (Figure 7E,F). The high expression of *HbSEO3* and *HbSEO4* in stem, leaf, bark, and latex suggested these genes might be related to latex production.

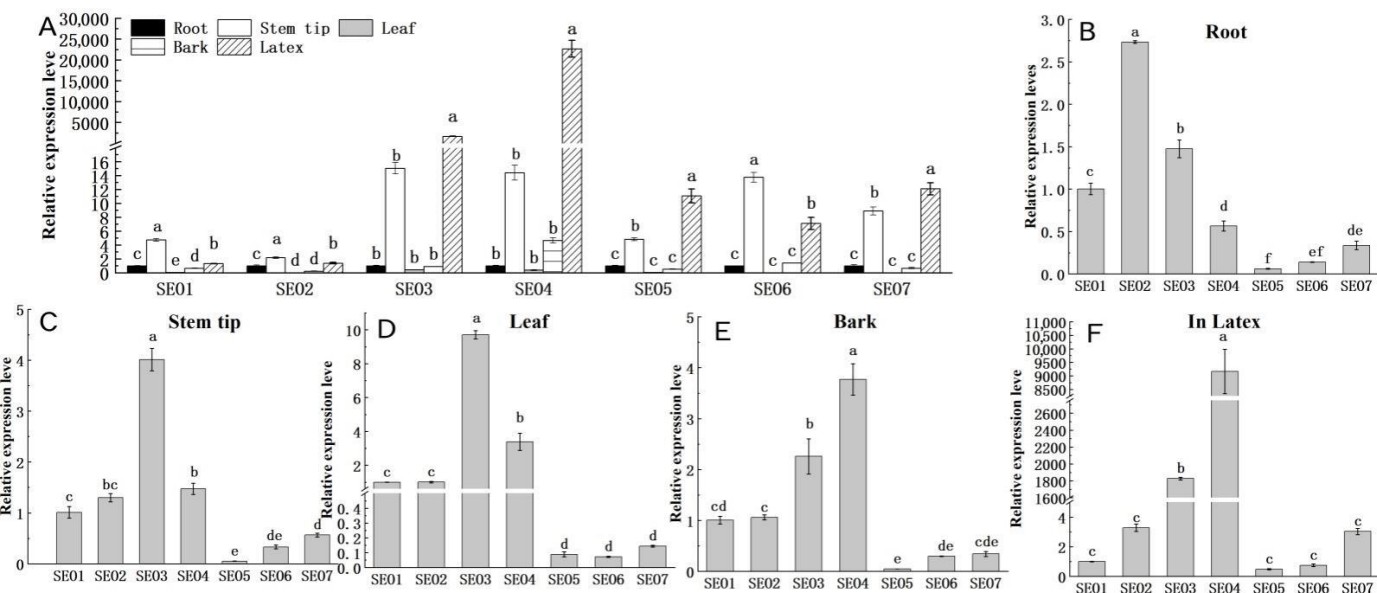

**Figure 7.** Relative expression of *HbSEOs* in different tissues and latex. (**A**) The expression of the same *HbSEOs* in different tissues and latex. (**B**–**F**) The expression of different *HbSEOs* in the same tissue and latex. (**A**–**F**) were plotted by using the same set of data, whereas different normalization criteria were utilized. In (**A**), the expression level of each gene in roots was used as a standard to normalize and compare the gene expression level among various tissues and latex, while in (**B**–**F**) the expression of *HbSEO1* in the corresponding tissues and latex was used as a standard, to perform normalization and compare the relative expression of different genes in the same tissue or latex. The error bars denote SE from three separate replications, different small-case letters signify statistical differences between means following Duncan's one-way ANOVA ($p < 0.05$), the same applies to the following.

### 3.5.2. Expression Patterns of *HbSEOs* Subjected to Various Hormone Treatments

In order to investigate the response of the *HbSEOs* to different hormone treatments, the rubber tree saplings were subjected to MeJA, GA, ETH, and SA treatments. The qRT-PCR results (Figure 8) showed that *HbSEOs* had altered expression patterns in response to various hormones. Under a 0.005% MeJA treatment, the *HbSEO* transcripts reached their peak at 0.5 h, and then were markedly downregulated. When treated with 3 mmol/L GA, the transcripts of *HbSEO1*, −2, −5, −6, and −7 were downregulated, while *HbSEO3* and *HbSEO4* reached the highest level at 4 h and were then downregulated. In response to 1% ETH treatment, the relative expression of *HbSEO2* and *HbSEO5* was upregulated at 4–8 h and then recovered to the normal level; whereas *HbSEO3* and *HbSEO4* were upregulated at 0.5–8 h, and downregulated after 12 h. When subjected to 200 μmo/L SA treatment, the relative expression of all *HbSEOs* genes was downregulated, except that *HbSEO5* was upregulated at 4–8 h and downregulated at 24 and 48 h.

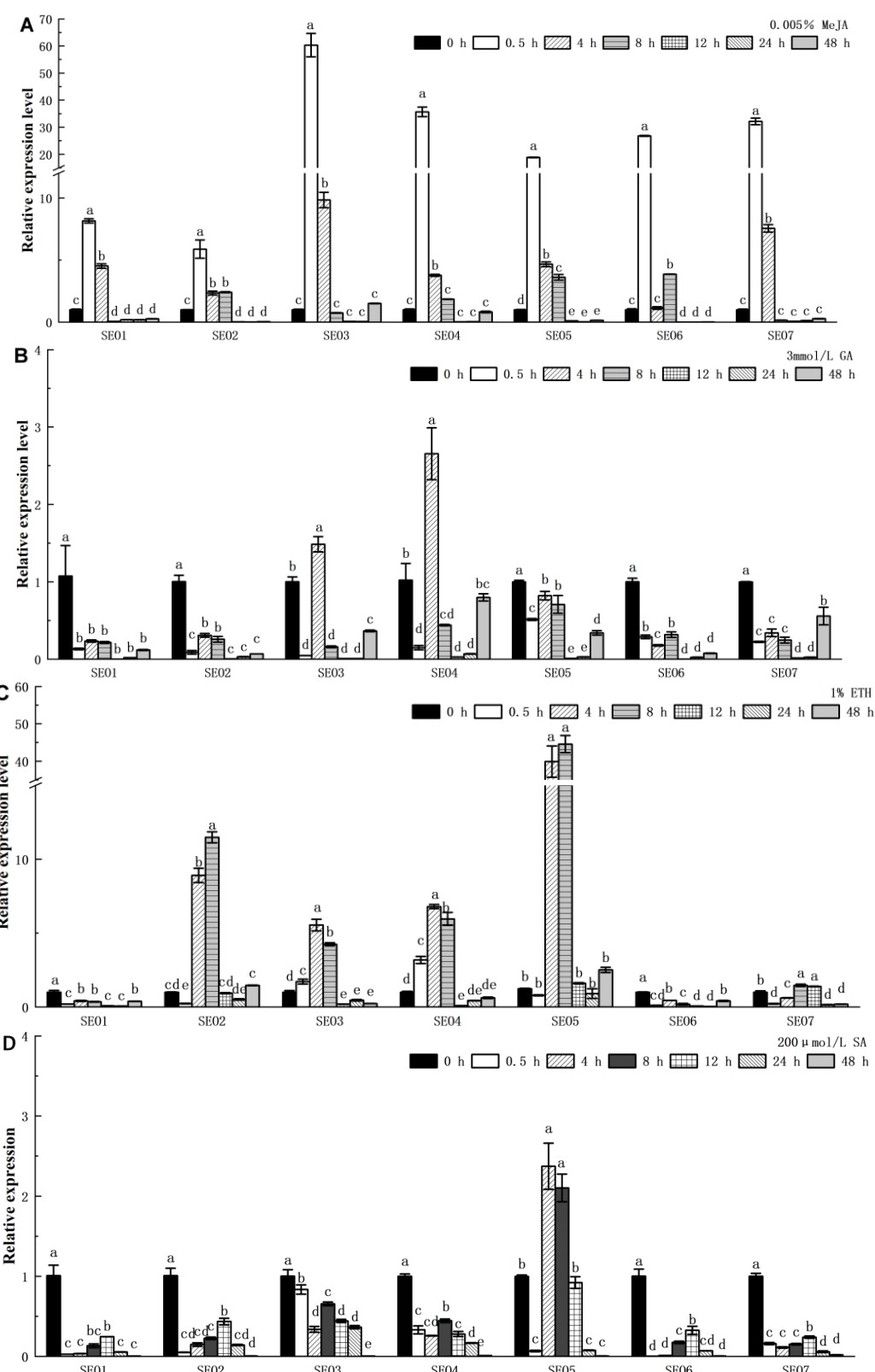

**Figure 8.** Relative expression profiles of *HbSEOs* under different hormone treatments. (**A**) 0.005% MeJA, (**B**) 3 mmol/L GA, (**C**) 1% ETH, and (**D**) 200 μmol/L SA. The rubber tree 18S gene was used as the internal control. To calculate the relative expression level of each gene in different treatment times, their expression level at 0 h was used to normalize the expression levels at other times. The relative expression of the target gene was calculated using the $2^{-\Delta\Delta CT}$ method. Different small-case letters represent statistical difference following Duncan's one-way ANOVA ($p < 0.05$) for same genes.

### 3.5.3. The Responses of *HbSEOs* to Mechanical Injury

After mechanical injury of the leaves, the *HbSEO1*, −2, −3, −4, −5, and −7 genes showed a similar expression pattern, which were all upregulated at first and then down-regulated (Figure 9). Among them, *HbSEO3* and *HbSEO4* showed the greatest increase in expression at 0.5 h, and then the relative expressions returned to that before treatment at 6 h. *HbSEO1*, −2, −5, and −7 were upregulated at 0.5–2 h and downregulated after 2 h. *HbSEO6* was downregulated at 0.5–1 h, then upregulated at 2 h, and downregulated after 2 h.

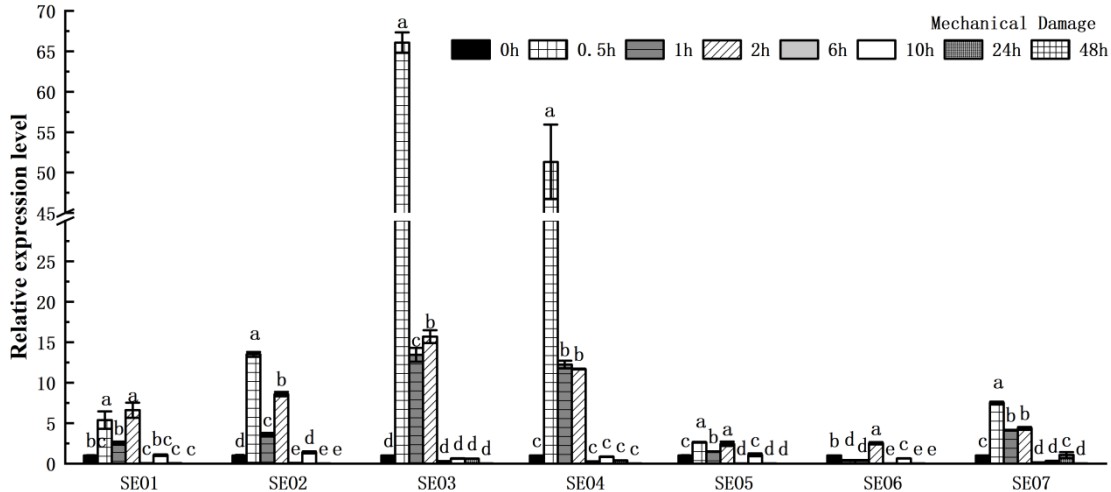

**Figure 9.** Relative expression variation of *HbSEOs* subjected to mechanical injury treatments. To calculate the relative expression level of each gene at different treatment times, the expression level of each gene in 0 h was used to normalize the expression levels at other times. Different small-case letters signify statistical differences between means following Duncan's one-way ANOVA ($p < 0.05$) for same genes for same genes.

### 3.5.4. Variation of *HbSEO* Genes Expressions in the Rubber Trees with Different Latex Flow Durations

The relative expressions of *HbSEOs* were compared in rubber tree barks and latices, with long (about 5 h) and short (about 1 h 50 min) latex flow durations (Figure 10). The results showed the gene relative expression of *HbSEO3* and *HbSEO4* in bark (Figure 10A) was higher than other genes. Moreover, the transcript abundances of *HbSEOs* were higher in the rubber trees with short latex flow than that with the long latex flow duration. In the latex (Figure 10B), the transcripts of *HbSEO3* and *HbSEO4* again showed the highest expression, and the relative gene expression of all *HbSEOs* was higher in the latex with short latex flow time.

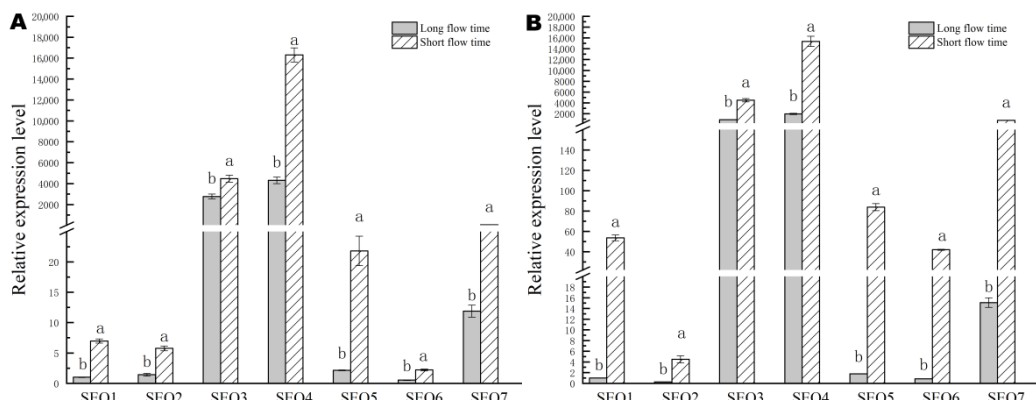

**Figure 10.** The relative expression of *HbSEOs* in the barks and latices of rubber trees with different flow durations. (**A**) in barks, (**B**) in latices. Data normalization was performed with the expression level of *HbSEO1* in the long duration; ANOVA was performed for the same gene with different flow times. Different small-case letters signify statistical differences between means following Duncan's one-way ANOVA ($p < 0.05$) for same genes.

## 4. Discussion

### 4.1. Genome-Wide Identification of HbSEO Gene Family and Their Classification

The *SEO* genes encoding forisomes relevant to sieve element blocking were first identified in *Medicago truncatula* L. and other legumes [16]. Later, they were reported in many non-Fabaceae plants having a similar function [8]. It is, therefore, suggested that *SEO* genes in angiosperms probably encode common P-proteins with a similar ultra-structure and function to Fabaceae forisomes [8]. All the *SEO* gene family members shared the following characteristics: (1) they were mainly expressed in phloem tissue and might be limited to sieve elements, (2), had conserved intron–exon gene structure, and (3) all the encoded proteins possessed three domains, i.e., SEO-NTD (SEO N-terminal domain), potential thioredoxin folding (Thioredoxin_like), and SEO-CTD (SEO C-terminal domain) [17]. Only when these three domains exist in combination, can the protein be considered a member of the *SEO* gene family. Based on the available data, the *SEO* family members varied, from three in *A. thaliana* (one of which is a pseudo gene) to 26 in soybean (*Glycine max* (L.) Merr.) [18]. Therefore, it was essential to perform a systematic survey of *SEO* members for rubber trees. From a genome-wide search, a total of seven HbSEO homologous genes were identified, which all featured the three domains (SEO-NTD, Thioredoxin_like, and SEO-CTD). The number of *SEO* genes in rubber trees is similar to the nine members reported in *P. trichcarpa* and *R. commounis*, eight members in *M. esculenta* (the species also belongs to Euphorbiaceae), and six members in *Solanum phureja* L.; and does not differ much from three members reported in *A. thaliana* and four members in *Mimulus guttatus* Fisch. ex DC. [16]. Previous research indicated that *P. trichcarpa* underwent one whole-genome triplication event (designated 'γ') and two doubling events. Although *A. thaliana* underwent the same γ event and one independent doubling event, massive gene loss and chromosomal rearrangement after genome duplication caused the arabidopsis genome to encode relatively less SEO genes [19–22]. Therefore, as a core eudicot plant, the rubber tree genome encoded a similar number of *SEOs* as other Euphorbiaceae plants, but more *SEOs* than *A. thaliana*.

Phylogenetic evolution analysis provides useful information for gene classification and function analysis. The SEO phylogenetic tree shows that the phylogenetic relationship between *HbSEOs* and cassava is the closest, and the seven *HbSEOs* and eight MeSEOs can be divided into two subgroups based on their evolutionary relationships and gene structures. Most SEOs from rubber trees and other plants were clustered into cluster I; and only HbSEO5 was clustered into cluster II, with the pseudogenes AtSEOc, and PtSEO5, RcSEO9, and MeSEO2. Further analysis of these proteins in cluster II showed that they all lack a motif at the 3′ end (Supplementary File S1). This result further supports the

phylogenetic classification and indicates that this motif might play a pivotal role in cluster II SEO proteins. Nevertheless, the functions of this motif and these proteins have never been studied, and this might provide a direction for the study of the protein function of this clade. In addition, HbSEO4 harbored one more motif4 than the other proteins, which implies that HbSEO4 might have different functions or expression patterns from the others. Further research is needed to confirm this fact.

### 4.2. Association of HbSEOs with Laticifer Plugging

The SEO proteins are accumulated in the cytoplasm of metabolically-active, undifferentiated sieve elements (SEs), and anchored to the plasma membrane when SEs mature [23]. After wounding damage, the SEO proteins detach from their parietal location and formed a gel-like mass to plug the downstream sieve plates, thereby preventing the loss of photoassimilates [24]. The gene expression pattern of *SEO* genes subjected to wounding and sealing has been reported in *M. truncatula*, *G. max*, *A. thaliana*, *S. phureja*, and *Nicotiana tabacum* L. It was found that, except for part of the *GmSEOs*, all *SEO* genes were significantly more highly expressed in the phloem-enriched tissues and played a crucial role in sieve element occlusion [8,25]. The rubber tree latex is the phloem cytoplasm flowing out after laticifers are severed. Nevertheless, the involvement of *HbSEOs* in rubber tree phloem laticifer plugging has not received much attention. In this study, a total of seven *HbSEOs* were identified from the rubber tree genome. Tissue specific expression analysis showed that the seven *HbSEO* genes were expressed in all the examined samples, including roots, stem tips, leaves, barks, and latex (Figure 7A); indicating that *HbSEOs* might be involved in rubber tree growth and development. Furthermore, the significant response of *HbSEOs* to mechanical injury, particularly *HbSEO3* and *HbSEO4* upregulation (Figure 9); and the significantly higher expression of *HbSEO*3 and *HbSEO*4 in leaf, bark, and latex than other genes (Figure 7A); and their relatively higher expression in latex than other tissues (Figure 7B), implied that these two genes might play important roles in laticifer plugging and latex yield. Therefore, further works regarding the involvement of *HbSEOs*, preferentially *HbSEO3* and *HbSEO4*, in the laticifer plugging and yield of latex are required.

Hormones such as GA, SA, ETH, and MeJA affected the growth and latex yield of rubber trees [4,26–31], among which ETH and MeJA are the most widely used stimulants to increase rubber tree latex production [30]. GA and SA increased rubber tree abiotic resistance [28,32] and exogenous MeJA could induce secondary laticifer differentiation; they could, therefore, increase rubber tree latex yield [33]. Nevertheless, ethylene and exogenous ethephon application increased latex yield by a different mechanism. They could prolong the latex flow time, by inhibiting blockage of the severed laticifer [28,29]. Mechanical wounding could also induce ethylene production in *Hevea* [29,34] and other plants [17]. *SEO* genes have been shown to be relevant to sieve element plugging after mechanical injury in other plants [8,25]. Therefore, it is speculated that *HbSEOs* might be directly involved in laticifer plugging and inducible by efficient rubber tree latex yield stimulants, such as ethephon. In response to exogenous ethephon application, *HbSEO* expression, particularly *HbSEO*3 and *HbSEO*4, was significantly upregulated at first and then downregulated after 12 h of application, which was consistent with the significant increase of latex flow duration and latex yield [35,36]. When surveying the promoter regions, all of the *HbSEOs* contained at least one ethylene cis-response element. Consequently, it is reasonable to assume that *HbSEOs* could be regulated by ethylene and associated with latex yield. The late down-regulation of *HbSEOs* by ethephon might lead to delayed laticifer blocking and increase in latex yield. Moreover, MeJA-responsive element, SA-responsive element, and GA-responsive element are also contained in the promoter of most *HbSEOs* (Figure 6); the transcription level of *HbSEOs*, therefore, changed by different degrees in response to MeJA, GA, and SA treatments, and ETH and MeJA were more effective in comparison to GA and SA (Figure 8). The regulatory effect of hormones and stresses to *HbSEOs* provide us with some useful clues for the selection an effective stimulant for rubber tree latex yield.

The laticifer plugging determined latex flow duration and was related to the latex yield [37–39]. When the latex flow time was longer, the latex yield was higher (Table 3). To further investigate the relationship between latex flow duration and *HbSEO* gene expression, the relative gene expressions of *HbSEOs* in rubber tree barks and latex were compared in a mature tapped rubber tree with varied latex flow durations (Table 3, Figure 10). It was shown that the *HbSEO* gene expressions in the bark and latex of trees with short flow duration were significantly higher than for the trees with long flow time. The higher expression of *HbSEOs* in short flow trees made them more susceptible to laticifer plugging, the latex flow duration and latex yield were, thus, decreased. These results further supported our hypothesis that *HbSEOs* are associated with laticifer plugging and latex yield.

**Table 3.** Rubber tree information used for different latex flow durations.

| Long Latex Flow Durations | | | | Short Latex Flow Durations | | | |
|---|---|---|---|---|---|---|---|
| Tree Number | Girth (cm) | Average Latex Flow Duration | Average Yield (mL·tree$^{-1}$·tapping$^{-1}$) | Tree Number | Girth (cm) | Average Latex Flow Duration | Average Yield (mL·tree$^{-1}$·tapping$^{-1}$) |
| 250 | 65 | 6 h 16 min | 394 | 59 | 64 | 1 h 57 min | 67 |
| 287 | 65 | 4 h 49 min | 212 | 293 | 64 | 1 h 48 min | 11 |
| 317 | 65 | 4 h 04 min | 192 | 301 | 67 | 2 h 07 min | 51 |
| 33 | 65 | 4 h 31 min | 342 | 271 | 63 | 1 h 48 min | 27 |
| 77 | 66 | 4 h 05 min | 206 | 333 | 65 | 1 h 54 min | 14 |
| Mean | 65.2 [A] | 4 h 45 min [A] | 269.2 [A] | | 64.6 [A] | 1 h 55 min [B] | 34 [B] |

Data were from 5 separate tapings. Means for long flow and short flow duration trees are given at the bottom of the table, different capital letters indicate significant difference between the means. Different capital letters for same parameter represent significant differences ($p < 0.01$).

## 5. Conclusions

Seven *HbSEOs* with complete domains were identified from the rubber tree genome and mapped onto four different chromosomes. Phylogenetic evolution analysis showed that *HbSEOs* can be divided into two subfamilies, according to their evolution relationship and motif structure. The promoters of these seven genes all contained cis-acting elements related to biotic and abiotic stresses, suggesting that *HbSEOs* may play an important role in regulating plant growth and development, and have a role in stress response. Based on the results of qRT-PCR analyses, the expression level of *HbSEOs* varied with tissues. The relatively high expression of *HbSEOs* in bark and latex, and their responses to mechanical damage and various hormones, implied their role in laticifer plugging and yield. This study is the first report of *SEO* genes in rubber trees. It provides a guide for the selection of *HbSEOs* for cloning and functional analysis, so as to unveil their roles in regulating rubber tree laticifer plugging and yield.

**Supplementary Materials:** The following supporting information can be downloaded at: https://www.mdpi.com/article/10.3390/f13030433/s1, File S1: Motif analysis of SEO proteins from other plants. Table S1: The SEO Protein ID (NCBI) of *Populus trichocarpa*, *Cassava*, *Arabidopsis*, *Ricinus communis*.

**Author Contributions:** F.A. and L.C. conceive and designed the experimental procedures; X.W., F.A. and W.P. conducted the experiments; X.W., F.A. and L.C. wrote the original manuscript; F.A., X.W., G.X. and Z.L. revised the manuscript. All authors have read and agreed to the published version of the manuscript.

**Funding:** This research was funded by funds from the Hainan Provincial Natural Science Foundation of China [321QN332], the Hainan Provincial Basic and Applied-basic Research Program (Natural Science) for High-level Talents [2019RC326], and the Earmarked Fund for China Agriculture Research System [CARS-33-ZP1]. And the APC was funded by [CARS-33-ZP1].

**Data Availability Statement:** The data and results are available to every reader upon reasonable request.

**Acknowledgments:** We thank the anonymous reviewers and editors for their valuable comments and revisions to improve the manuscript.

**Conflicts of Interest:** The authors declare no conflict of interest.

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
