# Peer review of "Identification of Sieve Element Occlusion Gene (SEOs) Family in Rubber Trees (Hevea brasiliensis Muell. Arg.) Provides Insights to the Mechanism of Laticifer Plugging"

_forests, doi:10.3390/f13030433_

Round 1

Reviewer 1 Report

The authors studied SEO genes in Rubber species and their roles in latex yield and laticifer plugging. These genes were compared to those in other species, and the promoter of Rubber SOEs were analysed for cis regulatory elements. Potential genes that are overexpressed when various treatments are applied to the Rubber tree are reported, which can be used to guide the selection of SOEs for cloning and functional analysis.

  • Figure 2 depicts the evolutionary relationship of Rubber SEO genes with other related species. Instead of using a clustering algorithm such as Neighbor Joining (NJ), which is not the most reliable, the Maximum Likelihood (ML) method for tree construction can be used. This will improve our understanding of gene evolutionary relationships and clustering.
  • The number of members in the SEO gene family ranges from three in Arabidopsis to twenty-six in Soybean. The loss or gain of this gene in rubber, as well as the divergence of this gene in other species studied, can be better understood if a divergence time tree is included.
  • Figure 6 shows an analysis of SEO promoters. The details of identified cis elements is not mentioned in the manuscript.
  • The gene HbSEO5 was reported in a cluster with AtSEOR2, which was assumed to be a pseudogene. When treated with different hormones, gene expression analysis revealed that this gene is highly upregulated at 4-8 h compared to other genes (Figure 8). A synteny analysis can aid in understanding the divergence of these genes from those found in other species.
  • The authors performed various plant treatments as well as gene expression analysis using qRTPCR. HbSEO3 and HbSEO4 were identified to have significant expression, implying that these genes may be related to latex production. Is there a correlation between these upregulated genes and their promoter regulatory elements when compared to other genes?

Reviewer 2 Report

The manuscript “Identification of sieve element occlusion genes (SEOs) family in rubber trees (Hevea brasiliensis) provides insights to the mechanism of laticifer plugging” by Xuan Wang, Linlin Cheng, Wentao Peng, Guishui Xie, Zifan Liu, Feng An, concerns very basic analysis of Hevea brasiliensis genes coding for P proteins associated with the blockage of sieve tubes after injuryIn my opinion, the manuscript is interesting for the Forest Journal because it concerns a plant of industrial importance and the analyzed genes participate in a variety of biological processes, including plant response to stresses and pathogens.

The manuscript requires a linguistic correction. I have highlighted regions of the manuscript or sentences that require linguistic correction, and in some cases have made suggestions for corrections.  Some of the chapters in the Results contain errors that need to be corrected. My main remarks concern:

1 – Paragraph  3.5.1. Tissue-specific expressions of HbSEOs.

There are differences in the relative expression value for individual genes between Figures 7A and 7B.

2 – Paragraph 3.5.2. Expression patterns of HbSEOs subjected to various hormone treatments.

Fig. 8 A - C are missing!

3 - Paragraph 3.5.3. The responses of HbSEOs to mechanical injury.

There are differences in relative expression in leaves for the HbSEO1, HbSEO2, HbSEO3, and HbSEO4 genes between Fig. 7 and Fig. 9.

4 – Discussion.

I cannot agree with the authors' statement that the presence of one pseudogene in given clade means that other genes in this clade are also pseudogens. Moreover, since SEO proteins from other plants (PtSEO5, RcSEO9 and MeSEO2) lie between HbSEO5 and AtSEOc, hence the authors' claim means that all genes in the clade “Cluster II”, genes between HbSEO5 and AtSEOc, are non-functional, see Fig. 2.

The rest of my comments on the manuscript are in “forest-1610076-peer-review-peer-review-JP.pdf”.

In conclusion, please make the corrections and linguistic proofreading.

Round 2

Reviewer 1 Report

I am satified with the response to referees' comments. I would recommend the current-revised version of the manuscript for publication.